# Tumor Tissue Microbiota in Colorectal Cancer: PCR Profile of FFPE Blocks in Associations with Metastatic Status

**DOI:** 10.3390/cells14191508

**Published:** 2025-09-27

**Authors:** Nikolay K. Shakhpazyan, Liudmila M. Mikhaleva, Nikolay K. Sadykhov, Konstantin Y. Midiber, Anton S. Buchaka, Zarina V. Gioeva, Alexander I. Mikhalev, Arkady L. Bedzhanyan

**Affiliations:** 1Avtsyn Research Institute of Human Morphology, Petrovsky National Research Center of Surgery, Moscow 119435, Russia; 2Institute of Medicine, Peoples’ Friendship University of Russia Named After Patrice Lumumba, 6 Miklukho-Maklaya St., Moscow 117198, Russia; 3Department of Hospital Surgery No. 2, Pirogov Russian National Research Medical University, Moscow 117997, Russia; 4Department of Abdominal Surgery and Oncology II (Coloproctology and Uro-Gynecology), Petrovsky National Research Center of Surgery, Moscow 119435, Russia

**Keywords:** colorectal cancer, formalin-fixed paraffin-embedded, PCR, tumor microbiota, *Fusobacterium nucleatum*, *Acinetobacter*, metastasis

## Abstract

Background: Tumor-associated microbiota are implicated in colorectal cancer (CRC). Formalin-fixed paraffin-embedded (FFPE) tumor tissue is widely available yet seldom profiled for microbiota. We tested whether quantitative and presence/absence signals of selected taxa in FFPE tumors associate with clinicopathological features. Methods: DNA from FFPE primary CRCs (*n* = 52) was assayed by a targeted PCR panel quantifying 30 bacterial taxa, *Candida* spp., and total bacterial load. Presence/absence combinations were selected by the Apriori algorithm with Fisher’s exact testing and 10,000-permutation empirical *p*-values. Quantitative features were modeled by LASSO logistic regression; discrimination of single taxa and combinations was evaluated by ROC/AUC. Results: In relative-abundance analyses, *Fusobacterium nucleatum* showed pro-metastatic value (AUC = 0.622). The best absolute-abundance model for metastasis combined *F. nucleatum*, *Faecalibacterium prausnitzii*, total bacterial load, and *Akkermansia muciniphila* (AUC = 0.739). Anti-metastatic directionality in relative-abundance models was driven by *Acinetobacter* spp.; the two-taxon set *Eubacterium rectale* + *Acinetobacter* spp. achieved AUC = 0.747. Conclusions: PCR-based profiling of FFPE CRC tumors is feasible and reveals hypothesis-generating patterns. Signals linking *F. nucleatum* to metastatic CRC and *Acinetobacter* spp. to non-metastatic disease merit validation in larger cohorts; tumor-tissue microbiome features may complement clinicopathological assessment.

## 1. Introduction

The colonic epithelium is a barrier whose cells exhibit a high proliferation rate and are exposed to a vast number of endogenous and exogenous carcinogenic factors. Consequently, it is not surprising that colorectal cancer (CRC) is one of the most common types of malignant tumors in adults. One of the key challenges is the prevention of this disease. According to the literature, primary prevention may include lifestyle modification, increased physical activity, and dietary interventions [1]. Secondary prevention of CRC involves early disease detection through various methods, including instrumental approaches (e.g., colonoscopy) and laboratory in vitro tests [2]. At last, tertiary prevention aims to improve treatment outcomes and the condition of patients with established CRC by dietary correction, modification of daily physical activity, elimination of harmful habits, and low-dose aspirin [3]. Each type of prevention currently requires further improvement, and modulation of the gut microbiota is considered one of the promising approaches.

The contents of the large intestine constitute the most densely populated bacterial ecosystem, which significantly influences the condition of the intestinal mucosa. It is well established that the composition of the microbiota affects processes such as epithelial barrier function, modulation of innate immunity, and prevention of pathogenic colonization—protective mechanisms that become impaired in epithelial tumors of the colon and rectum [4,5]. The state of the microbiota plays a significant role in CRC pathogenesis [6,7], with inflammation being a particularly influential factor in the initiation and progression of CRC [8,9]. Inflammatory activity is, in turn, modulated by dysbiotic processes [10]. Therefore, investigating the role of specific microorganisms and their combinations, as well as identifying simple and effective diagnostic approaches for detecting qualitative and quantitative shifts in the microbiota, may help establish individualized microbiome-targeted strategies. Such strategies could enhance all levels of prevention—tumor development prevention, detection of pro-carcinogenic tendencies, and improvement of treatment outcomes in CRC.

The aim of this study was to determine the association between microbiota composition—both qualitative and quantitative—identified in DNA samples extracted from tumors with various pathological and morphological characteristics of CRC patients, based on PCR analysis.

## 2. Materials and Methods

### 2.1. Characteristics of Patients with Colorectal Cancer (CRC)

The study included 52 patients with colorectal cancer (26 males and 26 females), aged 32–78 years (mean age, 60.1 years; median, 63.5 years). The tumor was predominantly localized in the left colon in 44 patients (84.6%), and in the right colon in 8 patients (15.4%).

The analyzed material corresponded to epithelial malignant tumors—adenocarcinomas. Tumor grading was performed using a three-tier system: well-differentiated tumors (G1) were identified in 44 patients (84.6%), moderately differentiated (G2) in 4 patients (7.7%), and poorly differentiated (G3) in 4 patients (7.7%).

Data on the tumor stage according to the TNM classification system (8th edition, 2017) were available for 40 patients. The most common finding was T3 tumors with different combinations of N (degree of spread to regional lymph nodes) and M (presence of distant metastasis) categories, including both locally confined forms (without tumor extension beyond the bowel wall) in 10 patients, and advanced forms (with extension beyond the bowel wall, including lymph node metastases) in 30 patients, among which distant metastases (M) were present in 21. In 12 patients, TNM stage data were unavailable.

Immunohistochemistry (IHC) was used to determine the mismatch repair (MMR) status of tumors. Primary antibodies were applied against MSH6 (Rabbit, clone EPR3945, Abcam, Cambridge, UK), MSH2 (Mouse, clone 25D12, DBS, Pleasanton, CA, USA), PMS2 (Mouse, clone MOR4G, Novocastra Leica, Vista, CA, USA), and MLH1 (Mouse, clone ES05, Novocastra Leica, Vista, CA, USA). MMR-proficient tumors were observed in 49 patients (94.2%), while MMR-deficient tumors were detected in 3 patients (5.8%).

The mutational profile of KRAS, NRAS, and BRAF genes was determined by real-time PCR using commercial reagent kits (TestGen, Moscow, Russia: TEST-NRAS-TKAN, TEST-KRAS-TKAN, TEST-BRAF-TKAN), according to the manufacturer’s instructions. The mutation panel included: KRAS: G12S, G12R, G12C, G12D, G12A, G12V, G13D; NRAS: Q61K, G12D, G12C, Q61L, G13D, Q61R, G13R, G12S; BRAF: V600E.

No mutations were identified in 39 patients (75.0%). Mutations in the KRAS gene were detected in 8 patients (15.4%), and BRAF mutations in 5 patients (9.6%).

### 2.2. Assessment of Quantitative and Qualitative Composition of the Microbiota by PCR

DNA was extracted from formalin-fixed paraffin-embedded (FFPE) blocks of primary colorectal tumors. Genomic DNA was extracted using the DNA-Tissue-M kit (TestGen, Moscow, Russia) according to the manufacturer’s protocol. In brief, 10 µm FFPE sections were deparaffinized by xylene-ethanol (ErgoProduction, Moscow, Russia) washes, and tissue lysis was performed in a proteinase K-containing buffer (TestGen, Moscow, Russia). Following lysis, the lysate was incubated for 1 hour at 90 °C to reverse formalin-induced cross-links; DNA was then captured on magnetic beads, washed, and eluted in TE buffer (TestGen, Moscow, Russia). DNA quality control was assessed spectrophotometrically (A260/A280 ≥ 1.7). At the PCR stage, a sample was considered acceptable if the total bacterial load was ≥10^5^, which serves as the endogenous internal control for PCR in this assay.

The quantitative status of the microbiota in DNA samples was determined by PCR using the “Kolonoflor Premium” kit (Alfa-Lab, Saint Petersburg, Russia) according to the manufacturer’s instructions. The microbial panel of the kit “Kolonoflor Premium” included the following taxa (30 bacterial taxons and 1 fungal genus, Candida): *Lactobacillus* spp., *Bifidobacterium* spp., *Escherichia coli*, *Bacteroides* spp., *Faecalibacterium prausnitzii*, *Bacteroides thetaiotaomicron*, *Akkermansia muciniphila*, *Enterococcus* spp., enteropathogenic *Escherichia coli*, *Klebsiella pneumoniae*, *Klebsiella oxytoca*, *Candida* spp., *Staphylococcus aureus*, *Clostridium difficile*, *Clostridium perfringens*, *Proteus vulgaris/mirabilis*, *Citrobacter* spp., *Enterobacter* spp., *Fusobacterium nucleatum*, *Parvimonas micra*, *Salmonella* spp., *Shigella* spp., *Blautia* spp., *Acinetobacter* spp., *Eubacterium rectale*, *Streptococcus* spp., *Roseburia inulinivorans*, *Prevotella* spp., *Methanobrevibacter smithii*, *Methanosphaera stadmanae*, *Ruminococcus* spp., as well as the total bacterial load.

The detected microorganisms and the total bacterial load were quantified as absolute values (expressed as the decimal logarithm (Lg) of the taxon copy number per PCR reaction). In addition, the taxa were normalized to the total bacterial load, expressed as the decimal logarithm of the relative proportion of each taxon.

### 2.3. Statistical Data Analysis

Statistical analyses were performed using Python v.3.12.

Descriptive statistics of the study groups and pairwise comparisons were performed with the pandas, NumPy, SciPy, and statsmodels libraries. Normality of distribution was assessed using the Shapiro–Wilk test. Parametric statistics were applied using ANOVA for multiple-group comparisons and the *t*-test for pairwise comparisons. Non-parametric statistics included the Kruskal–Wallis test for multiple groups and the Mann–Whitney U-test for pairwise comparisons. For multiple testing correction, the Benjamini–Hochberg method was applied. For groups with a small number of non-zero values (≤3), statistical significance was assessed using Fisher’s exact test for larger groups used Pearson χ^2^ test.

To identify combinations of taxa in the presence/absence mode (qualitative test), associated with specific clinical features, the Apriori algorithm (mlxtend.frequent_patterns.apriori) was applied using a binary matrix of taxa presence/absence. For each identified combination of taxa (1 to 3 taxa), a 2 × 2 contingency table was constructed with the target group. Statistical significance was evaluated using Fisher’s exact test (scipy.stats.fisher_exact) for small sample sizes, or Pearson’s χ^2^ test (scipy.stats.chi2_contingency) for larger ones. To increase the robustness of significance estimation, a permutation test with 10,000 permutations was performed for each taxa combination. At each permutation, the *p*-value was recalculated using Fisher’s exact test or χ^2^ test. The empirical *p*-value was defined as the proportion of permutations in which the *p*-value was less than or equal to the original value. The following metrics were calculated for each taxa combination: support (frequency of occurrence of the combination), confidence (proportion of the target group among cases with the combination), lift (ratio of observed to expected probability of the rule), the original *p*-value (Fisher’s exact or χ^2^ test), and the empirical *p*-value from the permutation test.

To evaluate the quantitative contribution of each taxon, L1-regularized logistic regression (LASSO) was applied (scikit-learn, LogisticRegressionCV), with five-fold cross-validation and optimization of the regularization parameter based on the maximum area under the ROC curve (ROC AUC). Prior to model building, quantitative variables were standardized using z-transformation (StandardScaler).

Model performance was evaluated by ROC curve analysis with calculation of ROC AUC. Statistical testing of differences in individual taxa between the positive and negative groups was conducted using the two-sided Mann–Whitney U-test. Correction for multiple testing was performed using the Benjamini–Hochberg method. LASSO results were used to identify significant taxa showing consistent directional associations with the target group. The selected taxa were further used to assess the predictive value of taxa combinations.

For taxa previously selected by LASSO, the diagnostic value of all possible combinations of 1 to 4 variables was evaluated. The classification ability of taxa combinations was assessed by ROC AUC. For each taxon within a combination, statistical testing was performed using the Mann–Whitney U-test. For each combination length (1, 2, 3, or 4 taxa), correction of the obtained *p*-values was performed separately using the Benjamini–Hochberg procedure to control the false discovery rate (FDR).

Heatmap construction was performed in R v.4.4.2 (RStudio Build 513) using the packages readxl, tidyverse, janitor, and pheatmap.

### 2.4. Ethical Approval and Informed Consent Statements

The study was conducted in accordance with the principles of the 1975 Helsinki Declaration and its 2013 version. The research protocol №8 was approved by the local ethical committee of the Petrovsky National Research Center of Surgery on 23 May 2025. All study participants provided written informed consent.

## 3. Results

Total bacterial load was detected in all tumor samples, with a mean (± SD) of 9.39 ± 0.42 log10 genome equivalents per PCR reaction. In addition, the following taxa were detected across samples: *Lactobacillus* spp., *Bifidobacterium* spp., *Escherichia coli*, *Bacteroides* spp., *Faecalibacterium prausnitzii*, *Bacteroides thetaiotaomicron*, *Akkermansia muciniphila*, *Klebsiella pneumoniae*, *Staphylococcus aureus*, *Clostridium difficile*, *Clostridium perfringens*, *Enterobacter* spp., *Fusobacterium nucleatum*, *Blautia* spp., *Acinetobacter* spp., *Streptococcus* spp., *Eubacterium rectale*, *Roseburia inulinivorans*, *Prevotella* spp., *Methanobrevibacter smithii*, *and Ruminococcus* spp.

None of the 52 cases showed detection of *Enterococcus* spp., enteropathogenic *Escherichia coli*, *Klebsiella oxytoca*, *Candida* spp., *Proteus vulgaris/mirabilis*, *Citrobacter* spp., *Parvimonas micra, Salmonella* spp., *Shigella* spp., or *Methanosphaera stadtmanae*. These taxa were excluded from all subsequent analyses and were not included in FDR correction procedures.

Descriptive statistics and pairwise comparisons were performed for predefined clinicopathological groupings: tumor grade (G1 vs. G2–G3 (hereinafter referred to as lower- vs. higher-grade groups)), primary tumor localisation (right colon—from cecum to hepatic flexure vs. left colon—from distal to the hepatic flexure to rectum), presence vs. absence of distant metastases, MMR deficiency status, presence vs. absence of KRAS/NRAS/BRAF mutations, and local extent of disease (intramural/locally confined vs. extramural extension).

Absolute and relative abundances of the analyzed taxa in tumor grade groups and presence/absence of distant metastases are presented in Figure 1.

Several trends in microbiome composition were observed across these groups. In the higher-grade group (G2–G3), Eubacterium rectale was detected in 1 of 6 patients (16.67%); the median was 0 (IQR 0–0; mean ± SD: 1.05 ± 2.57 log10 units), whereas in the G1 group the bacterium was not detected (0 of 44). The unadjusted Mann–Whitney U-test yielded *p* = 0.008, whereas Fisher’s exact test did not indicate significance (*p* = 0.12).

An increase in Fusobacterium nucleatum levels was observed in patients with metastatic disease compared with those without distant metastases. In the metastatic group, F. nucleatum was detected in 6 of 21 cases (28.6%); the median was 0 (IQR 0–5.3; mean ± SD: 1.85 ± 3.01 log10 units). In the non-metastatic group, detection occurred in 1 of 19 cases (5.3%); the median was 0 (IQR 0–0; mean ± SD: 0.28 ± 1.22 log10 units). The unadjusted Mann–Whitney U-test gave *p* = 0.047, whereas Fisher’s exact test yielded *p* = 0.094.

Thus, based on descriptive statistics and pairwise comparisons, there were possible tendencies toward detection of Eubacterium rectale in higher-grade tumors (G2–G3) and elevated Fusobacterium nucleatum in metastatic cases. However, these differences did not achieve statistical significance with the current analytical approach (unadjusted *p*-values used for trend exploration).

To assess the significance of different combinations of microorganisms, we applied the Apriori algorithm in a presence/absence mode, binarizing taxa as detected vs. not detected. The most informative combinations were observed when stratifying tumor samples from patients with distant metastases. The analysis used a minimum support threshold of support > 0.10. For each identified association, standard Apriori metrics were computed: support, confidence, and lift. Statistical significance was evaluated using Fisher’s exact test (*p*-value) and a permutation test with 10,000 permutations (empirical *p*-value). Although the significance threshold (*p* < 0.05) was not reached, the top taxon combinations exhibited elevated lift values (up to 1.63), suggesting a potential association between the corresponding bacterial taxa and metastatic status (Table 1).

Notably, *Fusobacterium nucleatum* appeared in all taxon combinations that exhibited the highest lift and confidence values. Moreover, all combinations involving *F. nucleatum* had identical support, confidence, and lift metrics, indicating that these combinations are fully nested within the same subset of patients. This pattern may reflect a dominant contribution of *F. nucleatum* to microbiological profiles associated with metastasis in this cohort, which is consistent with literature reports on its role in CRC progression. Nevertheless, given the lack of statistically significant differences, this observation requires validation in an expanded cohort.

To assess associations between individual taxa, their combinations in quantitative form and clinicopathological groupings, we performed logistic regression with L1 regularization (LASSO) to evaluate the discriminative importance of microorganisms. The results are presented in Figure 2.

As a result of the LASSO analysis, we selected sets of microorganisms exhibiting consistent prognostic directionality with respect to the clinicopathological groups. The analysis was performed separately for absolute abundances (log10 units per PCR reaction, denoted as Lg[taxon]) and for relative measures computed as log10(total bacterial load) − log10(taxon), reflecting each taxon’s proportion relative to the total bacterial load. Taxa with concordant directionality (pro-metastatic, anti-metastatic, associated with higher grade, associated with lower grade, associated with mutations, or associated with wild-type status) were carried forward for downstream analyses to identify the most informative combinations of taxa for discriminating clinicopathological groups of CRC patients (Table 2).

To select taxon combinations with prognostic value and to assess statistical significance, we performed ROC analysis. As a result, we identified significant combinations of microorganisms whose quantitative measurement—based on the absolute abundances of the detected taxa (log10 genome equivalents per PCR reaction)—may discriminate metastatic colorectal cancer (Table 3).

Each combination includes *Fusobacterium nucleatum*; moreover, *F. nucleatum* alone exhibits prognostic value, suggesting that this taxon may be a principal determinant for the metastatic CRC group. The most informative combination appears to include *Fusobacterium nucleatum*, *Faecalibacterium prausnitzii*, total bacterial load, and *Akkermansia muciniphila* (area under the ROC curve, AUC = 0.739), see Figure 3.

In the analysis of relative abundances, *Fusobacterium nucleatum* demonstrated pro-metastatic prognostic value (AUC = 0.622; *p* = 0.047), whereas other combinations were not significant.

Notably, ROC analysis of taxa associated with anti-metastatic, based on absolute abundances did not reach significance, but significant combinations emerged when evaluating relative abundances (Table 4).

Acinetobacter spp. appears in every combination; moreover, *Acinetobacter* spp. alone exhibits prognostic value, suggesting that this taxon may be a key determinant for the non-metastatic CRC group. The most informative combination appears to include *Eubacterium rectale* and *Acinetobacter* spp. (area under the ROC curve, AUC = 0.747), see Figure 4.

No significant associations were identified in the taxon groups associated with tumor grade or with KRAS/NRAS/BRAF mutational status. The top-performing combinations for the mutation-negative (wild-type) group included *Lactobacillus* spp., but none reached statistical significance (*p* = 0.13).

## 4. Discussion

It is now well established that the gastrointestinal microbiota influences the development and course of disease, including chronic inflammatory conditions and malignancies of the colon and rectum.

In our descriptive statistics and pairwise group comparisons, we observed a trend toward detection of *Eubacterium rectale* in tumors of grade G2 or higher. The role of *Eubacterium rectale* (also known as *Agathobacter rectale*) in CRC remains controversial. On the one hand, this microorganism is a major producer of short-chain fatty acids which exerts anti-inflammatory effects. Accordingly, CRC-associated microbiota is often depleted of this species [11,12]. On the other hand, there is evidence that butyrate may promote CRC growth by potentiating M2 macrophage polarization and by activating Treg and Breg cells, i.e., anti-inflammatory actions that may become pro-carcinogenic once a tumor is established [13]. *E. rectale* can potentiate inflammatory carcinogenesis in the murine colon via an E. rectale endotoxin effect on the NF-κB signaling pathway [14].

As widely reported in connection with CRC, *Fusobacterium nucleatum* already emerged at the level of descriptive statistics and pairwise comparisons in our data. Association between *F. nucleatum* and metastatic stages, was confirmed by Apriori-based association mining and LASSO logistic regression with ROC analysis. It have shown that *F. nucleatum* can persist within distant CRC metastases [15]. High *F. nucleatum* burden has been associated with poor prognosis and increased metastasis [16,17].

We found that a combination comprising *F. nucleatum*, *Faecalibacterium prausnitzii*, total bacterial load, and *Akkermansia muciniphila* modestly improved the prognostic model compared with assessment of *F. nucleatum* alone. *F. prausnitzii* is generally regarded as exerting strong protective, antitumor effects as a butyrate producer; its relative abundance is reduced in CRC, and it modulates immune responses with anti-inflammatory actions [18,19,20]. But it was already mentioned that butyrate can activate pro-carcinogenic effects in established tumors [13]. The association we observed with *F. nucleatum* may therefore reflect dysbiosis or compensatory shifts within the microbiota and warrants further study.

The literature does not provide a clear, direct link between total bacterial load and CRC. Although some studies use bacteria—regardless of taxon—as a biomarker (e.g., levels of circulating bacterial DNA in blood [21]). A possible explanation for our observed association is increased bacterial adhesion to tumor tissue.

*Akkermansia muciniphila* is also considered to act predominantly through protective mechanisms, including suppression of the AHR/β-catenin signaling axis [22]. Notably, activation of the β-catenin pathway is one of the key mechanisms implicated in the carcinogenic effects of F. nucleatum [23]. The association we observed may thus represent a compensatory response of the microbiota to *F. nucleatum*.

An interesting observation is the association of *Acinetobacter* spp. with the absence of distant metastases, which in our analysis emerged when evaluating relative abundances. The genus *Acinetobacter* is not generally characterized as strongly protective; there are reports linking Acinetobacter to CRC, whereas other studies associate it more with adenomas than with CRC [24,25]. It has also been reported that *Acinetobacter* is infrequently detected specifically within CRC tumor tissue [26]. At present, there is no compelling evidence that *Acinetobacter* spp. directly drives CRC development or progression. Changes in its abundance may reflect other processes within the colonic microbiome.

Of note, LASSO analysis indicated that *Lactobacillus* spp. abundance was associated with the absence of mutations activating the MAPK pathway (KRAS, NRAS, BRAF). Our study did not find a statistically significant association between *Lactobacillus* spp. and KRAS/NRAS/BRAF mutations. *Lactobacillus* spp. are often described as having antitumor properties [27]. While no association of *Lactobacillus* spp. with KRAS/NRAS/BRAF has been consistently reported, correlations with other frequent CRC alterations—such as PIK3CA mutations that activate the PI3K–AKT pathway—have been described [28].

Our study demonstrates the feasibility of microbiome assessment using a simple, widely accessible approach—PCR applied to the most commonly available oncologic biomaterial, formalin-fixed paraffin-embedded (FFPE) histological blocks. The microbiota detected in our work appear to be strongly associated with tumor tissue, potentially representing a mucosa-associated component residing within crypts and interacting closely with the colonic mucosa. Notably, the most influential taxon in our dataset—*F. nucleatum*—exhibits strong cellular adhesion mediated by the Fap2 adhesin, which may favor its preservation during histological processing [29].

This study has several limitations. First, the sample size (*n* = 52) constrains statistical power and increases the risk of overfitting in multivariable models (LASSO). The reported AUCs should be viewed as potentially optimistic and in need of external validation in planning further research. Second, our targeted PCR panel (30 taxa) covers only a small fraction of the gut microbiome (although enriched with significant for assessing the biocenosis of the large intestine). Accordingly, the findings are considered as hypothesis-generating. Third, this is a single-center cohort with clinicopathologic heterogeneity (site/stage), the data matrix is sparse (many zeros), which may affect the robustness of associations. Finally, despite multiple-testing control and sensitivity analyses, independent validation and broader, untargeted profiling in larger cohorts are required to confirm the observed signals and to quantify their incremental prognostic value over standard clinicopathologic assessment.

## 5. Conclusions

Our study demonstrates the feasibility of qualitative and quantitative microbiota analysis in histological (FFPE) material by PCR. We observed trends suggesting differential relevance of *F. nucleatum* for metastatic CRC and of the genus *Acinetobacter* for non-metastatic CRC. Characterization of the microbiota in tumor tissue may have diagnostic value and could inform strategies for microbiome modulation in CRC patients for therapeutic or preventive purposes after thorough validation of the microbial panel and clinical testing of the developed diagnostic algorithm.

## Figures and Tables

**Figure 1 cells-14-01508-f001:**
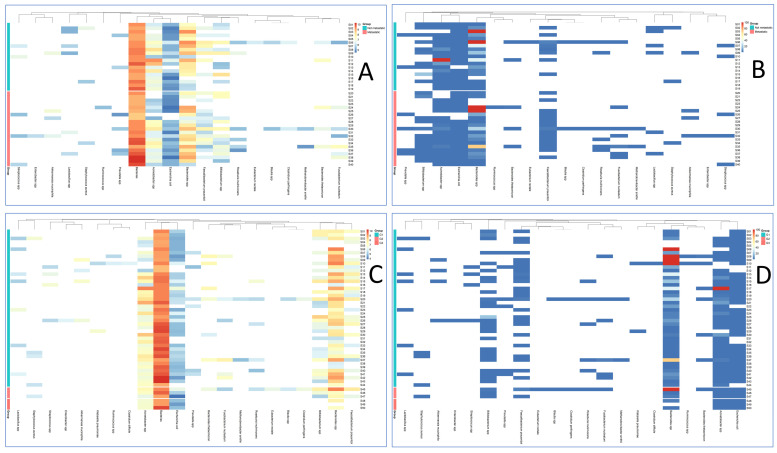
Heatmap of taxon abundance stratified by the presence of distant metastases (**A**,**B**) and by tumor grade (**C**,**D**). Data are shown as absolute values (decimal logarithm, log10, of taxon genome equivalents per PCR reaction; panels (**A**,**C**)) and as percentages relative to the total bacterial load (panels (**B**,**D**)). White indicates absence of the taxon (bacterial DNA not detected). By visual inspection, hierarchical clustering reveals a subset of metastatic cases with higher absolute Fusobacterium nucleatum signals (**A**), whereas after normalization the relative contribution of Acinetobacter increases predominantly in non-metastatic samples (**B**). Many taxa exhibit zero-inflated, sparse patterns (white blocks). Grade-stratified panels (**C**,**D**) show less coherent structure with only modest shifts in several commensals (e.g., Eubacterium rectale). These patterns are descriptive; formal statistics are reported in the Results.

**Figure 2 cells-14-01508-f002:**
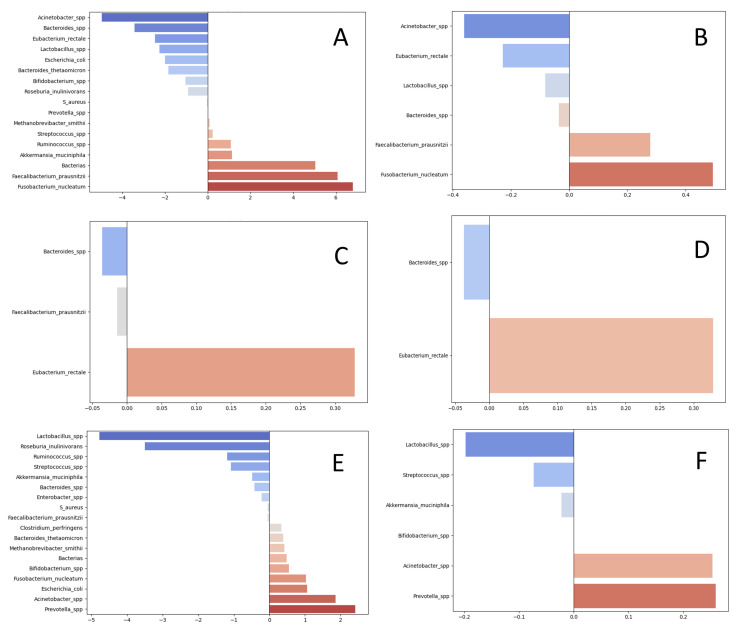
LASSO-derived taxon coefficients. Coefficients for the investigated taxa that discriminate patients across clinicopathological groups. A positive coefficient indicates a taxon positively associated with (i.e., “pro” for) the target group; a negative coefficient indicates a taxon inversely associated with (i.e., “contra” to) the target group. Both absolute abundance (log10 genome equivalents per PCR reaction) and relative abundance (percentage of the total bacterial load) were analyzed. Panels: (**A**,**B**) metastatic CRC group (absolute and relative abundance, respectively); (**C**,**D**) higher-grade CRC group (G2 and above; absolute and relative, respectively); (**E**,**F**) CRC with detected KRAS, NRAS, or BRAF mutations (absolute and relative, respectively).

**Figure 3 cells-14-01508-f003:**
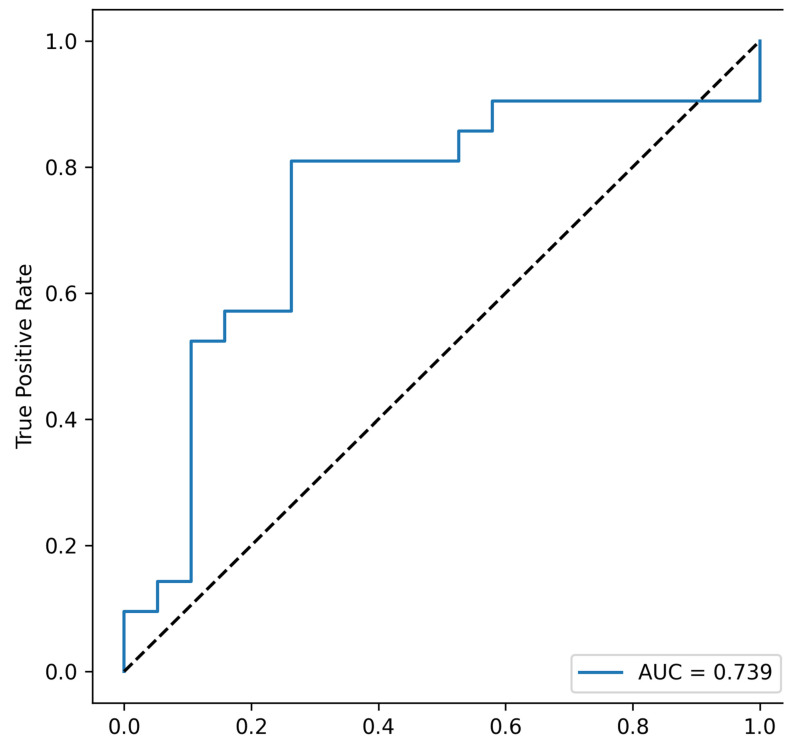
Receiver operating characteristic (ROC) curve for the four-taxon combination—Fusobacterium nucleatum, Faecalibacterium prausnitzii, total bacterial load, and Akkermansia muciniphila—in discriminating metastatic CRC. This combination of taxa showed the highest area under the curve (AUC), which may indicate the greatest predictive potential for persons with CRR metastases.

**Figure 4 cells-14-01508-f004:**
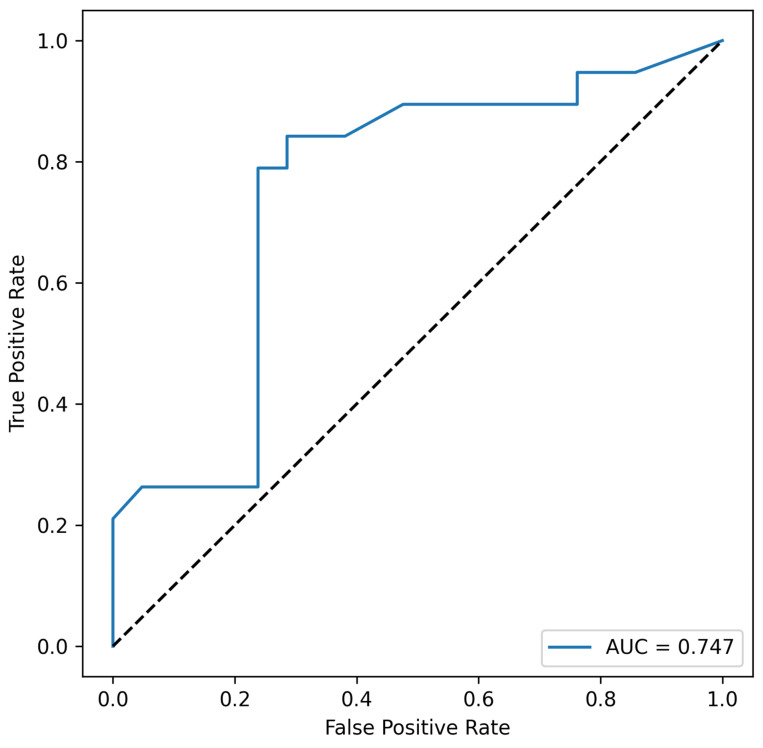
ROC-curve of taxa combination «*Eubacterium rectale*, *Acinetobacter* spp.» of non-metastatic CRR. This combination of taxa showed the highest area under the curve (AUC), which may indicate the greatest predictive potential for persons without CRR metastases.

**Table 1 cells-14-01508-t001:** The most valuable taxon combinations (Apriori algorithm).

Taxon Combinations	Support	Confidence	Lift	*p*-Value	Empirical *p*
*Bacteroides* spp., *Fusobacterium nucleatum*	0.175	0.29	1.632653	0.095	0.094
*Bacteroides* spp., *Acinetobacter* spp., *Fusobacterium nucleatum*	0.175	0.29	1.632653	0.095	0.094
*Acinetobacter* spp., *Fusobacterium nucleatum*	0.175	0.29	1.632653	0.095	0.094
*Escherichia coli*, *Acinetobacter* spp., *Fusobacterium nucleatum*	0.175	0.29	1.63	0.095	0.094
*Fusobacterium nucleatum*	0.175	0.29	1.63	0.095	0.094
*Escherichia coli*, *Fusobacterium nucleatum*	0.175	0.29	1.63	0.095	0.094
*Bacteroides* spp., *Escherichia coli*, *Fusobacterium nucleatum*	0.175	0.29	1.63	0.095	0.094

**Table 2 cells-14-01508-t002:** Taxa concordant directionality for absolute abundances and relative measures (LASSO analysis).

Groups	Taxons	Regression Coefficients for Absolute Abundances (LASSO)	Regression Coefficients for Relative Measures (LASSO)
pro-metastatic	*Fusobacterium nucleatum*	6.79	0.49
	*Faecalibacterium prausnitzii*	6.07	0.27
	Total bacterial load	5.03	0
	*Akkermansia muciniphila*	1.14	0
	*Ruminococcus* spp.	1.08	0
	*Streptococcus* spp.	0.23	0
	*Methanobrevibacter smithii*	0.09	0
	*Prevotella* spp.	0.01	0
anti-metastatic	*Acinetobacter* spp.	−4.95	−0.36
	*Bacteroides* spp.	−3.43	−0.04
	*Eubacterium rectale*	−2.47	−0.23
	*Lactobacillus* spp.	−2.27	−0.08
	*Escherichia coli*	−2.00	0
	*Bacteroides thetaomicron*	−1.84	0
	*Bifidobacterium* spp.	−1.04	0
	*Roseburia inulinivorans*	−0.91	0
	*Staphylococcus aureus*	−0.03	0
associated with higher grade	*Eubacterium rectale*	0.33	0.33
associated with lower grade	*Bacteroides* spp.	−0.04	−0.04
	*Faecalibacterium prausnitzii*	−0.01	
associated with mutations	*Prevotella* spp.	2.41	0.26
	*Acinetobacter* spp.	1.86	0.25
	*Escherichia coli*	1.06	0
	*Fusobacterium nucleatum*	1.03	0
	*Bifidobacterium* spp.	0.54	0
	Total bacterial load	0.48	0
	*Methanobrevibacter smithii*	0.42	0
	*Bacteroides thetaomicron*	0.39	0
	*Clostridium perfringens*	0.34	0
associated with wild-type status	*Lactobacillus*_spp.	−4.77	−0.2
	*Roseburia_inulinivorans*	−3.5	0
	*Ruminococcus*_spp.	−1.19	0
	*Streptococcus*_spp.	−1.08	−0.07
	*Akkermansia_muciniphila*	−0.48	−0.02
	*Bacteroides*_spp.	−0.42	0
	*Enterobacter*_spp.	−0.22	0
	*S_aureus*	−0.05	0
	*Faecalibacterium_prausnitzii*	−0.04	0

**Table 3 cells-14-01508-t003:** Receiver operating characteristic (ROC) analysis of absolute abundances taxa combinations discriminating metastatic colorectal cancer.

Combination	AUC (Area Under Curve)	*p* Value
*Fusobacterium_nucleatum, Faecalibacterium_prausnitzii*, Total bacterial load, *Akkermansia_muciniphila*	0.739	0.047
*Fusobacterium_nucleatum, Faecalibacterium_prausnitzii*, Total bacterial load, *Ruminococcus*_spp.	0.721805	0.047
*Fusobacterium_nucleatum*, *Faecalibacterium_prausnitzii*, *Akkermansia_muciniphila*, *Ruminococcus*_spp.	0.699248	0.047
*Fusobacterium_nucleatum*, Total bacterial load, *Akkermansia_muciniphila*, *Ruminococcus*_spp.	0.677945	0.047
*Fusobacterium_nucleatum, Faecalibacterium_prausnitzii*, Total bacterial load	0.709273	0.047
*Fusobacterium_nucleatum, Faecalibacterium_prausnitzii, Akkermansia_muciniphila*	0.694236	0.047
*Fusobacterium_nucleatum, Faecalibacterium_prausnitzii, Ruminococcus*_spp.	0.674185	0.047
*Fusobacterium_nucleatum*, Total bacterial load, *Akkermansia_muciniphila*	0.672932	0.047
*Fusobacterium_nucleatum,* Total bacterial load, *Ruminococcus*_spp.	0.610276	0.047
*Fusobacterium_nucleatum, Akkermansia_muciniphila, Ruminococcus*_spp.	0.645363	0.047
*Fusobacterium_nucleatum, Faecalibacterium_prausnitzii*	0.671679	0.047
*Fusobacterium_nucleatum*, Total bacterial load	0.607769	0.047
*Fusobacterium_nucleatum, Akkermansia_muciniphila*	0.642857	0.047
*Fusobacterium_nucleatum, Ruminococcus*_spp.	0.622807	0.047
*Fusobacterium_nucleatum*	0.622807	0.047

**Table 4 cells-14-01508-t004:** ROC analysis of relative abundances of taxa in persons without CRR metastases.

Combination	AUC	min_*p*_Value
*Eubacterium_rectale*, *Acinetobacter*_spp.	0.747	0.0104
*Acinetobacter*_spp.	0.738	0.0104
*Lactobacillus*_spp., *Eubacterium_rectale*, *Acinetobacter*_spp.	0.724	0.0104
*Lactobacillus*_spp., *Acinetobacter*_spp.	0.711	0.0104
*Bacteroides*_spp., *Lactobacillus*_spp., *Eubacterium_rectale*, *Acinetobacter*_spp.	0.642	0.0104
*Bacteroides*_spp., *Eubacterium_rectale*, *Acinetobacter*_spp.	0.634	0.0104
*Bacteroides*_spp., *Lactobacillus*_spp., *Acinetobacter*_spp.	0.632	0.0104
*Bacteroides*_spp., *Acinetobacter*_spp.	0.629	0.0104

## Data Availability

All data and materials are available upon reasonable request. Ad-dress the request to N.K.S. (Nikolay K. Shakhpazyan) (email: nshakhpazyan@gmail.com) Avtsyn Research Institute of Human Morphology, Petrovsky National Research Center of Surgery, Moscow 119435, Russia.

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
