# Peer review of "Tumor Tissue Microbiota in Colorectal Cancer: PCR Profile of FFPE Blocks in Associations with Metastatic Status"

_cells, 2025, doi:10.3390/cells14191508_

Round 1
Reviewer 1 Report
Comments and Suggestions for Authors
This manuscript presents an original and timely investigation of tumor-associated microbiota in colorectal cancer using PCR-based profiling of FFPE tumor tissues. The study addresses an important research gap, as FFPE blocks are widely available but underutilized for microbiome characterization. The authors employ a rigorous methodology, including a targeted PCR panel, multiple statistical approaches (descriptive analysis, Apriori algorithm, LASSO regression, ROC/AUC analysis), and careful interpretation of findings.
- Several sentences are too long and contain multiple ideas (e.g., Introduction lines 34–42). Breaking them down would improve readability.
- Some terms are inconsistently capitalized (e.g., “Bacterias” in Table 2 should be “Total bacterial load” or “Bacteria”).
- Provide more detail on DNA extraction quality controls (e.g., DNA yield, integrity assessment).
- Patient characteristics section is clear, but percentages should be checked: e.g., “44 patients; 84.6%” localized in left colon is inconsistent (44/52 = 84.6% but right colon is 7/52 = 13.5%; totals to 51, not 52). Clarify missing data.
- Figures and tables: legends need more descriptive captions. For example, Figure 1 caption should describe what patterns were observed (not just the variables plotted).
- The discussion is strong but could be more concise. Some sections repeat Introduction material (e.g., roles of F. nucleatum).
- Table 2 mixes English and Russian (“Общая бактериальная масса”). Translate consistently into English.
- Sample size (n = 52) is relatively small for LASSO and ROC models with multiple variables. This limitation should be acknowledged more directlyin discussion
Author Response
Comments 1: Several sentences are too long and contain multiple ideas (e.g., Introduction lines 34–42). Breaking them down would improve readability.
Response: Thank you for the comment. Several long sentences were rephrased to make them shorter.
Comments 2: Some terms are inconsistently capitalized (e.g., “Bacterias” in Table 2 should be “Total bacterial load” or “Bacteria”).
Response: Thank you for pointing this out. “Bacterias” were changed to “Total bacterial load” in Tables.
Comments 3: Provide more detail on DNA extraction quality controls (e.g., DNA yield, integrity assessment).
Response: Thank you for the comment. We have added DNA extraction quality details to the Methods. Briefly, DNA was extracted from FFPE according to the manufacturer’s protocol; quality was assessed spectrophotometrically (A260/A280 ≥ 1.7). Sample adequacy and lack of inhibition were confirmed at the PCR stage by total bacterial load ≥ 10^5, which serves as an endogenous internal control.
Comments 4: Patient characteristics section is clear, but percentages should be checked: e.g., “44 patients; 84.6%” localized in left colon is inconsistent (44/52 = 84.6% but right colon is 7/52 = 13.5%; totals to 51, not 52). Clarify missing data.
Response: Thank you for the comment. We re-audited the source clinical records, resolved discrepancies, and corrected the patient characteristics.
Comments 5: Figures and tables: legends need more descriptive captions. For example, Figure 1 caption should describe what patterns were observed (not just the variables plotted).
Response: Thank you for the comment. We have expanded the captions for Figures 1, 3, and 4 and clarified the title of Table 3.
Comments 6: The discussion is strong but could be more concise. Some sections repeat Introduction material (e.g., roles of F. nucleatum).
Response: The discussion was shortened - some sentences were shortened, and repetitions about the need for further validation were removed (a separate paragraph, "Limitations" was created for this purpose).
Comments 7: Table 2 mixes English and Russian (“Общая бактериальная масса”). Translate consistently into English.
Response: Thank you for your comment. Corrected.
Comments 8: Sample size (n = 52) is relatively small for LASSO and ROC models with multiple variables. This limitation should be acknowledged more directly in discussion
Response: Thank you for your comment. A paragraph "Limitations" in the Discussion was created for this purpose.

Reviewer 2 Report
Comments and Suggestions for Authors
General points
This is an interesting and well-presented paper. The aim of the Shakhpazyan et al study was to determine if there was an association between microbial presence identified in DNA extracted from formalin preserved colorectal tumour tissue with various pathological characteristics, based on microbial PCR analysis. The methodology followed was appropriate but in the results and conclusion there is too much interpretation of the data from a relatively small sample size of only 50 or so tumour samples.
Specific points
‘a targeted PCR panel quantifying 30 bacterial taxa.
What if the most important microbial species in terms of prognosis of tumour progression were not covered by the 30 microbial taxa PCR panel screened ? There are an indeterminate but still much more than 30 taxonomic groups within the human gut microbiome.
‘Signals linking F.nucleatum to metastatic CRC and Acinetobacter spp. to non-metastatic disease merit validation in larger cohorts; tumor-tissue microbiome features may complement clinicopathological assessment.’
This is a single study with a diverse tumour range extracted, and a small patient sample size. Also, not all the gut microbial species that might be involved were covered in the PCR screen. Therefore, it is not really clinically appropriate to assign quite so much prognostic significance to the presence of any single species without more investigation.
Author Response
Comments 1. This is an interesting and well-presented paper. The aim of the Shakhpazyan et al study was to determine if there was an association between microbial presence identified in DNA extracted from formalin preserved colorectal tumour tissue with various pathological characteristics, based on microbial PCR analysis. The methodology followed was appropriate but in the results and conclusion there is too much interpretation of the data from a relatively small sample size of only 50 or so tumour samples.
Response Thank you for evaluating our work and for your constructive comments. Our study is hypothesis-generating and assesses the applicability of one of the most clinically practical approaches—PCR—for profiling the tumor-tissue microbiome. The limitations of this approach are outlined in the “Limitations” paragraph at the end of the Discussion.
Comments 2. ‘a targeted PCR panel quantifying 30 bacterial taxa. What if the most important microbial species in terms of prognosis of tumour progression were not covered by the 30 microbial taxa PCR panel screened ? There are an indeterminate but still much more than 30 taxonomic groups within the human gut microbiome.
Response. Thank you for the comment. Indeed, shotgun metagenomics typically detects ~100–200 microbial species per individual in the colon. Our PCR panel is designed for quantitative assessment of the colonic microbiocenosis—detecting obligate representatives (bifidobacteria, lactobacilli, E. coli) and opportunistic species. While not exhaustive, it enables feasibility assessment and hypothesis generation; this limitation is acknowledged in the “Limitations” section.
Comments 3. ‘Signals linking F.nucleatum to metastatic CRC and Acinetobacter spp. to non-metastatic disease merit validation in larger cohorts; tumor-tissue microbiome features may complement clinicopathological assessment.’ This is a single study with a diverse tumour range extracted, and a small patient sample size. Also, not all the gut microbial species that might be involved were covered in the PCR screen. Therefore, it is not really clinically appropriate to assign quite so much prognostic significance to the presence of any single species without more investigation.
Response. Thank you for the comment. As previously discussed, this aspect is addressed in the “Limitations” paragraph. In addition, the “Conclusions” section explicitly emphasizes the need for further validation and clinical evaluation of microbiome panel–based methods.
